# Prevention and Management of Iron Deficiency/Iron-Deficiency Anemia in Women: An Asian Expert Consensus

**DOI:** 10.3390/nu15143125

**Published:** 2023-07-13

**Authors:** Rishma Dhillon Pai, Yap Seng Chong, Lyra Ruth Clemente-Chua, Rima Irwinda, Trang Nguyen Khanh Huynh, Noroyono Wibowo, Maria Corazon Zaida Gamilla, Zaleha Abdullah Mahdy

**Affiliations:** 1Department of Obstetrics and Gynaecology, Lilavati Hospital, Mumbai 400050, India; rishmapai@hotmail.com; 2Department of Obstetrics and Gynaecology, Yong Loo Lin School of Medicine, National University of Singapore, Singapore 117597, Singapore; obgcys@nus.edu.sg; 3Department of Obstetrics and Gynecology, The Medical City, Pasig 1605, Philippines; 4Fetomaternal Division, Department of Obstetrics and Gynecology, Faculty of Medicine, University of Indonesia—Cipto Mangunkusumo Hospital, Jakarta 10430, Indonesia; rima.irwinda@yahoo.com (R.I.); wibowonoroyono@yahoo.com (N.W.); 5Department of Obstetrics and Gynecology, Pham Ngoc Thach University of Medicine, Ho Chi Minh 700000, Vietnam; tranghnk08@gmail.com; 6Department of Obstetrics and Gynecology, University of Santo Tomas Hospital, Manila 1008, Philippines; zngamilla_md@yahoo.com; 7Department of Obstetrics and Gynaecology, Faculty of Medicine, Universiti Kebangsaan Malaysia, Kuala Lumpur 56000, Malaysia

**Keywords:** iron deficiency, iron deficiency anemia, women, Delphi, consensus

## Abstract

The lack of standardized clinical practice impeding the optimal management of iron deficiency (ID) and iron deficiency anemia (IDA) in women is a global concern, particularly in the Asia-Pacific region. The aim of this study was to determine best practices through a Delphi consensus process. In Round 1, panelists were asked to rate their level of agreement with 99 statements across four domains: identification, diagnosis and assessment, prevention, and treatment of ID/IDA in women. In Round 2, panelists reappraised their ratings in view of the collective feedback and responses to Round 1. After two rounds, consensus (≥85% agreement) was reached for 84% of the Delphi statements. Experts agreed on the role of presenting symptoms and risk factors in prompting assessments of anemia and iron status in women. Experts repeatedly called for prevention, recommending preventive iron supplementation for pregnant women irrespective of anemia prevalence levels, and for non-pregnant adult women, adolescent girls, and perimenopausal women living in areas with a high prevalence of anemia. Experts unanimously agreed to prescribing oral ferrous iron as first-line therapy for uncomplicated ID/IDA. The recommendations and clinical pathway algorithms generated should be used to inform clinical practice and standardize the care of women at risk or presenting with ID/IDA in the Asia-Pacific region.

## 1. Introduction

Iron deficiency (ID) is one of the most common micronutrient deficiencies that disproportionately affects females throughout their lifecycle via menstruation (blood loss), pregnancy (fetal demands), and bleeding in childbirth [1,2,3]. Iron deficiency occurs as a spectrum, ranging from iron depletion without anemia to impaired erythropoiesis and anemia [4]. In 2019, anemia affected 1.8 billion people worldwide; ID is the predominant cause [5]. Of concern, this estimate does not begin to account for the many cases of ID in the absence of anemia [6]. Females, especially those in developing regions, are particularly vulnerable to ID, and anemia due to ID is the leading cause of years lived with disability among women of reproductive age in low- and middle-income countries [3,5,7]. According to the World Health Organization (WHO) estimates for 2016, anemia was diagnosed in 32.5% of non-pregnant women, 40.1% of pregnant women, and 32.8% of women of reproductive age, with Southeast Asia being the region with the highest prevalence of anemia in all three subgroups of women [8].

Iron serves an integral role in key cellular processes, including energy metabolism, cell signaling, gene expression, immunity, and cell growth regulation and differentiation [1,4,9]. Therefore, it is important to recognize that symptomatic ID can occur without anemia [10]. The systemic consequences of ID in women are extensive and potentially serious if left untreated [4,7]. In addition to the well-known features of anemia, ID can lead to non-hematological symptoms, which have often been overlooked. These can include fatigue, reduced physical endurance, hair loss, brittle nails, defective structure or function of epithelial tissues, pica, restless legs syndrome, decreased cognitive performance, and behavioral disturbances [4]. ID in women impairs overall health and quality of life; it is associated with an increased risk of adverse maternal and neonatal outcomes [4].

The enduring burden of anemia in women suggests that ID in women is still inadequately assessed, diagnosed, and treated to prevent progression to iron deficiency anemia (IDA) [7,9]. In 2014, the WHO set a goal of reducing anemia in women of reproductive age by 50% by 2025; this and other global targets for improving maternal, infant, and young child nutrition have since been extended to 2030 [1,3,11]. The WHO recommends intermittent or daily iron and folic acid supplementation as a public health intervention to improve iron status and reduce the risk of anemia in women of reproductive age, depending on pregnancy and postpartum status, menstruation, population-level prevalence, and the diagnosis of chronic disease or infection [3,12].

Despite the progress made, the absence of clear clinical guidance that focuses on women has led to considerable variations in clinical practice and patient outcomes. Therefore, the aim of this Delphi consensus was twofold: (1) to generate high-quality expert consensus statements; and (2) to develop clinical pathway algorithms for the identification, assessment, prevention, and treatment of ID/IDA in four key subgroups of women (i.e., pregnant women, non-pregnant adult women, adolescent girls, and perimenopausal women) presenting to the obstetrics and gynecology clinic in the Asia-Pacific region.

## 2. Materials and Methods

### 2.1. Study Design and Expert Panel

This paper uses the term ‘Delphi consensus’, referring to a systematic process of gaining consensus through controlled feedback by a group of experts recognized for their knowledge and experience in the subject matter. The ‘Delphi technique’ is used to describe the steps of the Delphi method applied in the process. ‘Delphi rounds’ refers to the iterative rounds of survey questionnaires sent to the panel members.

In view of the limited high-quality evidence around ID/IDA prevention and management in women, we employed a two-round Delphi technique (Figure 1). The Delphi technique has been applied across a wide range of medical disciplines to establish consensus and guide clinical practice, including the field of ID [13,14,15,16,17]. The modified Delphi technique, an approach that requires an in-person or virtual meeting to finalize consensus [18], was not considered for this study because it was challenging to schedule a time when all panelists could participate. Furthermore, this was a deliberate decision to allow authors adequate time to review the relevant literature and results of each Delphi round.

For this Delphi consensus, we assembled an expert panel composed of eight leading clinician–researchers representing obstetrics, gynecology, and women’s health from India (n = 1), Indonesia (n = 2), Malaysia (n = 1), Philippines (n = 2), Vietnam (n = 1), and Singapore (n = 1). Members of the expert panel were selected based on the following criteria: a practicing obstetrician–gynecologist with more than 15 years of experience; a regionally or nationally recognized expert in obstetrics and gynecology; considerable experience in the diagnosis, evaluation, and management of ID/IDA; previous experience as an advisory board member; and an editorial board member for a reputable regional journal. The Delphi surveys were administered anonymously to the panel between February and September 2022 using SurveyMonkey (www.surveymonkey.co.uk, accessed on 2 January 2022). Notably, all panelists were asked to provide feedback based on their perspectives, experience, and the latest literature, with the goal of improving practices across the Asia-Pacific region.

### 2.2. Development of Items for Round 1 of the Delphi Survey

The statements for Delphi Round 1 were developed from a review of the literature. The PubMed database was searched for full-text English-language articles published between 2016 and 2021. For this search, a combination of the following keywords was used: iron deficiency anemia, pregnancy/pregnant women, non-pregnant women, adolescents/adolescent girls, perimenopause/perimenopausal women, prevention/prophylactic, treatment/therapeutic, iron status assessment/evaluation, serum ferritin/ferritin, TSAT/transferrin saturation, socioeconomic status/factors, cultural factors/practices, iron supplementation/therapy, iron dose/dosage, guideline, and consensus. De-duplication of the search results identified 1763 publications. Of these, 202 were classified as relevant based on their titles and abstracts, with 10 additional publications identified from Google searches. The expert panel had access to all relevant information and documents during the Delphi rounds and during the development of this manuscript.

Appendix A) present all statements included in Delphi Round 1 across four domains: identification; diagnosis and assessment; prevention; and the treatment of ID/IDA in women.

A six-point Likert scale was used for panelists to rate their level of agreement or disagreement with each Delphi item: strongly disagree, disagree, somewhat disagree, somewhat agree, agree, and strongly agree. The Delphi statements were derived from published research and recommendations; therefore, we expected a fairly high level of agreement and set an 85% threshold (the sum of agree and strongly agree) for both Rounds 1 and 2. A free-text field was included for each statement to allow panelists the opportunity to explain their responses and propose new statements for Round 2.

### 2.3. Development of Items for Round 2 of the Delphi Survey

The results of Round 1, including free-text comments and feedback, were tabulated to inform the development of the Round 2 survey. A copy of these results was also shared with panel members in Round 2, as they reconsidered their responses to statements that did not meet consensus in Round 1. Similar to Round 1, free-text fields were included in Round 2 for panelists to justify and elaborate their responses.

### 2.4. Grading of Statements

After two Delphi rounds, the level of agreement for each statement was graded using a scale that has previously been described [19]: grade U (100%; unanimous), grade A (90–99%; near unanimous), grade B (78–89%; strong agreement with little variance), and grade C (67–77%; moderate agreement). For this Delphi consensus, the grading system was modified to include grade D (0–66%) to indicate statements that reflected extremely diverse opinions.

### 2.5. Development of Clinical Pathway Algorithms

Based on the results of both Delphi rounds, statements were formatted into clinical pathway algorithms around pre-determined domains (i.e., identification, diagnosis and assessment, prevention, and treatment of ID/IDA), with graded recommendations assigned to each decision point.

## 3. Results

Appendix A present the results of Delphi Round 1 and Round 2. Table 1 shows the number of statements retained in each round. In Round 1, experts retained 72 (73%) of the initial 99 statements. The Delphi Round 2 survey removed two statements and included 16 new additional statements, in which experts retained 95 (84%) of 113 statements. Clinical pathway algorithms were developed based on key statements retained in the final round.

### 3.1. Identification of ID/IDA in Women

After two Delphi rounds, consensus was reached on the need for the early detection of ID in all targeted subgroups of women (i.e., pregnant women, non-pregnant adult women, adolescent girls, and perimenopausal women); however, experts did not identify non-pregnant adult women and perimenopausal women as ‘high-risk’ subgroups for ID and the development of IDA. Experts reached a positive consensus on the majority of statements relating to presenting symptoms and risk factors that should prompt assessment of anemia and iron status; the statement that remained controversial concerned the significance of clinical pallor in the detection of ID/IDA in pregnant women.

### 3.2. Diagnosis and Assessment of ID/IDA in Women

Experts agreed that lower-than-normal serum ferritin is the most specific early marker for the diagnosis of ID in women. A positive consensus was almost reached (75% agreement) on serum ferritin being the most effective diagnostic strategy for ID in pregnancy; however, no consensus was achieved concerning the serum ferritin threshold for the diagnosis of ID during the first trimester of pregnancy. Importantly, a positive consensus was reached on the need for serum ferritin testing in anemic women to confirm IDA. No consensus was reached on the role of serum inflammatory markers, clinical signs, and oral iron therapy in excluding or supporting a diagnosis of IDA. Appendix A illustrates the clinical pathway algorithm for the identification, and diagnosis and assessment of ID/IDA in women.

### 3.3. Prevention of ID/IDA in Women

All statements concerning the importance of preventing ID/IDA in women reached a positive consensus in the first Delphi round. Experts agreed that preventive iron supplementation should be recommended for all targeted subgroups of women residing in settings where screening for ID/IDA is not adequate. In addition, consensus was reached on the provision of preventive iron supplementation for pregnant women irrespective of anemia prevalence levels, and for other targeted subgroups of women living in areas with a high anemia prevalence. Almost all statements describing oral iron dosing recommendations for prevention of ID/IDA achieved a positive consensus in Round 2; however, the optimal oral iron dosing for pregnant women in settings with a low anemia prevalence remained controversial. The decision tree shown in Appendix A depicts the clinical pathway algorithm for the prevention of ID/IDA in women.

### 3.4. Treatment of ID/IDA in Women

Experts agreed that oral ferrous iron should be recommended as first-line therapy for all targeted subgroups of women with uncomplicated ID/IDA. Following two Delphi rounds, consensus was also reached on statements indicating oral iron dosing recommendations for the treatment of mild IDA, as well as those describing situations where intravenous iron is appropriate. Additional new statements regarding the combination of multiple micronutrients with oral iron reached a positive consensus. Statements that remained controversial concerned oral iron dosing recommendations for the treatment of mild-to-moderate IDA and the state of evidence guiding the optimal therapeutic dosing for moderate-to-severe ID/IDA in adolescent girls, and for severe IDA in non-pregnant adult women, adolescent girls, and perimenopausal women. Appendix A defines the clinical pathway algorithm for the treatment of ID/IDA in women.

## 4. Discussion

The overarching purpose of this expert consensus is to provide clinicians practicing within the Asia-Pacific region with a straightforward set of recommendations that reflect current practices and can be readily implemented to improve long-term health outcomes for women at risk or presenting with ID/IDA. Previously published Asia-Pacific regional expert recommendations were developed to reduce the morbidity and mortality associated with end-stage IDA in the pregnancy and postpartum phase, and were therefore not addressing the continuum of prevention to early intervention and treatment [20]. Additionally, the threshold guidance for the detection of ID/IDA is outdated and not consistent with the more recent WHO guidelines. In view of this and the widely observed practice variations in the region, statements generated for this Delphi consensus encompass the identification, diagnosis and assessment, prevention, and treatment of ID/IDA across key subgroups of women: pregnant women, non-pregnant adult women, adolescent girls, and perimenopausal women.

### 4.1. Identification of ID/IDA in Women

Relative to pregnant women and adolescent females who are widely recognized as two groups most vulnerable to ID/IDA [21,22,23,24], many clinicians do not identify adult and perimenopausal women as ‘high-risk’ despite the fact that approximately 40% of non-pregnant women have low iron stores [25]. In addition, many national and international guidelines recognize the importance of screening and treating ID/IDA associated with heavy menstrual bleeding (HMB), which is estimated to affect 18–38% of women of reproductive age, and is higher among perimenopausal women [26].

Although ID is a leading cause of anemia, ID, IDA, and anemia should not be conflated [6,27]. The largest amount of circulating iron (~80%) is used for hemoglobin (Hb) synthesis; therefore, the absence of anemia is assumed to imply adequate iron stores [9,28]. However, IDA is the result of ID, as Hb synthesis is typically preserved until the advanced stages of ID [9].

Therefore, ID among women is easily missed, leaving early stages of treatable ID unidentified and untreated [27]. For this reason, risk factors and presenting symptoms indicative of ID assume an important role in the diagnostic process, although certain symptoms of ID do not always manifest until IDA develops. When symptoms are present in adolescent females with non-anemic ID, they commonly include fatigue, and decreased verbal learning and memory [29]. In perimenopausal women, these may include fatigue and impaired exercise capacity [2]. However, the extent to which these non-hematological symptoms are manifested before anemia remains unclear, and given the lack of published evidence to directly inform recommendations, there are no firm pathways to guide their investigations or monitoring [30,31].

Pallor of the skin, conjunctivae, and nail bed is one of the most recognized signs of IDA [32,33]. The quantitative assessment of pallor for the diagnosis of anemia remains an area of active investigation, with a wide range of estimated sensitivities and specificities [34,35]. Although it cannot be relied on to diagnose IDA, the presence of pallor in women should prompt the clinician to assess anemia and iron status, and consider the diagnosis of IDA, especially when its presence is a change from normal in the patient [32,36]. In cases of non-anemic ID, non-hematological symptoms, such as fatigue, can be present without anemic Hb levels.

HMB is a well-established risk factor for ID/IDA in pregnant and menstruating females, who often normalize or conceal their bleeding due to negative cultural perceptions of bleeding [1,2]. In addition to heavy uterine bleeding, multiparity increases the risk of ID during pregnancy and should always prompt testing and management, if ID/IDA is suspected [37]. Accordingly, a recent prospective cohort study showed that Hb lower than 12.0 g/dL in the first trimester of pregnancy was predictive of IDA at the end of pregnancy [38]. Identified risk factors for ID/IDA post-delivery include postpartum blood loss (>500 mL), uncorrected anemia detected during pregnancy, and symptoms suggestive of ID/IDA [33].

### 4.2. Diagnosis and Assessment of ID/IDA in Women

The WHO defines a diagnostic threshold for ID based on serum ferritin (SF) and severity threshold for anemia based on Hb [27]. Supported by a low to very low certainty of evidence, an SF cutoff of <15 μg/L is recommended for otherwise healthy adolescents, non-pregnant adults, and older adults [27,39]. This cutoff value is increased to <70 μg/L in the context of inflammation or infection [39]. The WHO also recommends an SF cutoff of <15 μg/L for pregnant women in their first trimester, but with no thresholds proposed for the second or third trimester of pregnancy [27,39]. Recent guidance by the British Society for Haematology (BSH) and the American College of Obstetricians and Gynecologists proposed a higher threshold of <30 μg/L to indicate ID during pregnancy, based on a 90% probability that iron stores are depleted when SF falls below 30 µg/L [27].

Determining an optimal threshold based on consensus methods is a significant challenge because there is no definitive evidence to support any of these recommended cutoffs [40]. A threshold of <15 μg/L would be deemed to be specific, but will likely miss many cases of ID [40]. A cutoff of <30 μg/L is highly sensitive, but can lead to more false positives. In pregnant women, a higher threshold is thought to address the uncertainty of iron consumption, proinflammatory changes, and the expansion of blood volume during pregnancy; however, this can result in increased health and economic burden of disease in countries where prevalence of ID/IDA is already very high [41]. As most Delphi panelists practice in resource-limited settings, it may come as no surprise that there was a unanimous consensus to implement an SF cutoff of <15 μg/L for the diagnosis of ID in otherwise healthy women.

Due to the acute phase nature of SF, serum inflammatory markers, such as ESR and CRP, are widely used in clinical practice to help interpret ferritin levels [42]. Markedly elevated ESR and CRP levels should prompt consideration of autoimmune disorders and chronic infections [42]. The WHO indicates that the concurrent assessment of inflammatory markers is only necessary in areas at high risk of infection or inflammation [39], which may explain the lack of consensus in considering inflammatory marker testing among Delphi panelists.

In most cases, IDA can be effectively diagnosed by assessing Hb and SF [9]. However, due to the lack of simple and inexpensive point-of-care tests, the practice of using the Hb concentration as a proxy for ID/IDA persists in many healthcare settings [27,43]. In anemic patients with suspected iron depletion states, a low SF provides a definitive diagnosis of IDA [27,44]. In cases with borderline SF, ID is typically confirmed with a panel of markers which may include serum iron, total iron-binding capacity, and transferrin saturation [44,45]. Newer iron status parameters, such as soluble transferrin receptors, reticulocyte Hb content, and serum hepcidin, are promising additions to the panel of diagnostic tests for ID; however, these assays are not widely available and are not adequately standardized [44].

A diagnosis of IDA in women should also prompt consideration of its cause, particularly gynecological bleeding [46]. Patients with obvious blood loss (e.g., women with menorrhagia) would not require an extensive diagnostic workup. In patients with IDA due to multiple mechanisms (e.g., the elderly, chronic inflammatory conditions, and malnutrition), further diagnostic testing may be necessary to determine its additional causes [47]. The evaluation and treatment of additional causes of IDA can aid in the resolution of IDA beyond iron therapy [47]. In iron-replete women with anemia, additional diagnostic tests can help determine the alternate causes of anemia and its appropriate intervention [47]. The BSH and the British Society of Gastroenterology (BSG) guidelines recommend that in anemic pregnant and non-pregnant adult women at low risk for non-adherence, a therapeutic trial of oral iron for 2–4 weeks can aid in diagnosing IDA [33,48].

### 4.3. Prevention of ID/IDA in Women

In the past, anemia prevention strategies for women had focused on preventing ID/IDA in pregnancy, which remains to be a leading cause of maternal deaths and adverse pregnancy outcomes in developing regions [49]. Additionally, it has been well established that women suffer from all consequences of ID/IDA whether or not they are pregnant. ID in non-pregnant women has been associated with reduced endurance and work performance, impaired intellectual and cognitive functions, and depression [50]. In addition, irritability, decreased endurance, and withdrawal behavior may be observed among women with mild ID [51].

Given its profound impact on aerobic and endurance capacity, IDA can result in reduced physical performance and work tolerance [50]. In addition to work productivity and voluntary activity, both ID and IDA have a clinically significant effect on fatigue and athletic performance in women [50]. Furthermore, iron adequacy during pregnancy can only be assured when the woman enters pregnancy with optimal iron reserves [52]. Therefore, clinicians should strive to prevent and treat even mild ID among adolescent females and non-pregnant adult women.

Extensive evidence supports the efficacy and safety of oral iron for the prevention and treatment of ID/IDA in both pregnant and non-pregnant women [7,53,54,55,56,57]. Current WHO guidance for the prevention of ID/IDA in women recommends daily supplementation of oral iron for 3 consecutive months in a year to all non-pregnant adult women and adolescent girls in countries with an anemia prevalence greater than 40% (i.e., 47% among reproductive-age and non-pregnant women in the WHO Southeast Asia Region [58,59]) and intermittent supplementation in settings with lower prevalence [12,24].

In pregnant women, the WHO advocates daily supplementation of oral iron and folic acid in settings where at least 40% of pregnant women are anemic (i.e., 48% among pregnant women in the WHO Southeast Asia Region [60]), and weekly intermittent doses where the prevalence is lower than 20%, and if oral iron is poorly tolerated [23]. This is consistent with the view that enterocytes become refractive to absorbing additional iron until they are replaced by new enterocytes after 5–6 days and, as such, the provision of iron doses at weekly intervals is expected to increase absorption and reduce gastrointestinal (GI) exposure to unabsorbed elemental iron [61]. In addition, the data show a 35% reduction in the incidence of anemia without an increased risk of GI adverse events [62].

Although the risks and benefits of preventive oral iron during pregnancy have been debated [38,63], experts in current Delphi consensus support the use of preventive oral iron in all targeted subgroups of women, including pregnant women, who reside in developing regions where access to ID/IDA screening is limited [7,63].

### 4.4. Treatment of ID/IDA in Women

The treatment of IDA should aim to replete iron stores and restore normal Hb levels [64]. Many expert groups and clinical guidelines recommend oral iron as first-line therapy for the treatment of uncomplicated ID/IDA in women [48,65,66,67]. As ferrous iron is more bioavailable and effective at restoring Hb levels compared with ferric iron, oral ferrous iron remains the standard treatment option for ID/IDA [67,68]. The three most widely used oral ferrous iron preparations are ferrous sulfate, ferrous gluconate, and ferrous fumarate (Table 2) [48,68,69].

In an effort to ensure a positive pregnancy experience for women, the WHO recommends increasing daily elemental iron from the standard daily antenatal dose of 30–60 mg to 120 mg for pregnant and postpartum women diagnosed with IDA, until Hb levels are normalized [23,71]. For non-pregnant adult women and adolescent girls with ID/IDA, clinicians are advised to follow national guidelines [24]; however, there is little published evidence and consensus on the recommended dose and dosing frequency of oral iron [23,67,72], resulting in inconsistencies in care and patient outcomes.

Although oral ferrous iron is conventionally dosed at 100–200 mg per day, given as three or four divided doses, experts in the present Delphi consensus agreed with the evolving evidence base that changing from consecutive-day to alternate-day schedules [61,73] and from divided to single morning doses [73] may optimize iron absorption and improve tolerability. Stoffel and colleagues found that single morning doses of 60–120 mg oral elemental iron plus ascorbic acid (i.e., a potent enhancer of iron absorption) on alternate days resulted in improved fractional iron absorption and may be the optimal oral dosing regimen for women with ID and mild IDA [61,67]. A single morning dose of oral iron augments plasma hepcidin, a negative regulator of systemic iron homeostasis. In view of the circadian increase in hepcidin over the day and to ensure that elevated hepcidin does not reach levels that would inhibit iron absorption, iron doses given in the afternoon or evening after a morning dose are not recommended [67,74].

There is a notion that patients who are given less frequent dosing may experience lower Hb increases, and thus may require longer durations of treatment [75]. Kaundal and colleagues demonstrated that while the median increase in Hb levels was faster in patients given two divided doses (i.e., 60 mg elemental iron BID) on consecutive days vs. single doses (i.e., 120 mg elemental iron QD) on alternate days, the difference in Hb levels as well as the total amount of iron was not observed at subsequent few weeks of treatment [76].

Although increasing evidence indicates similar or improved clinical response with alternate-day dosing [65,74], daily dosing of oral iron has maintained its role as a go-to treatment approach in many countries worldwide. In fact, recently published data from a large randomized controlled trial in India revealed that in 200 adults with IDA treated with either alternate-day or daily oral iron, there were no clinically meaningful differences in Hb improvement or safety outcomes between the two treatment arms [77].

However, the decision to prescribe alternate-day or consecutive-day oral iron therapy should depend on the severity of ID/IDA and patient preference (i.e., the desired rate of Hb response and tolerance to oral iron). If the rate of Hb response is not important but tolerance is, which is the case in mild IDA, alternate-day supplementation should be considered. In cases of moderate IDA requiring a rapid Hb response, consecutive-day supplementation remains a pertinent option, which may explain the high level of consensus achieved in the current Delphi consensus for statements relating to the role of daily oral iron in the treatment of women with mild to moderate IDA. After the correction of Hb, oral iron treatment should be continued for at least 3 months, and in clinical practice, it may be preferable to discontinue oral iron once normal SF levels are restored [7,67,78].

In addition to alternate-day dosing, adherence might improve with lower iron doses or selecting ferrous gluconate [75,79]. Notably, recent BSH guidelines recommend a lower range of 40–80 mg oral elemental iron every morning for the treatment of mild IDA. Additionally, newer iron formulations, such as sucrosomial iron, have a lower dose requirement compared with commonly available iron supplements [80]. While recent guidelines, including the BSH and BSG, recommend ferrous fumarate, sulphate, or gluconate for the treatment of ID/IDA, data from a systematic review assessing the tolerability of different oral iron formulations suggest improved adherence with ferrous gluconate or sulfate vs. ferrous fumarate [33,48,81]. Importantly, in clinical practice, it may be preferable to discuss with the patient to determine the optimal oral iron dosing, frequency, and duration, with the goal of optimizing efficacy and adherence [74].

In addition to iron, vitamin B12, folic acid, copper, and zinc are essential micronutrients for the synthesis of Hb [82]. Deficiency in any of these micronutrients can lead to anemia. Single micronutrient deficiencies (MNDs) rarely occur in humans [82,83], and MNDs are known to co-exist and interact. In patients with anemia, ID is more likely to be associated with other multiple MNDs [83,84]. Therefore, multiple-micronutrient (MMN) supplementation has been suggested to be more effective than iron alone in reducing anemia [83,84]. A Cochrane review provides compelling evidence that oral MMN supplementation during pregnancy is associated with improved birth outcomes (i.e., reduced risk of small-for-gestational age, low birthweight, and stillbirths) compared with oral iron (with or without folic acid), with no important benefit or harm on mortality outcomes [84]. Based on the latest evidence and considerations, the 2020 WHO guidance on antenatal care recommends oral MMN supplementation in settings where implementation research has been conducted to establish the positive impact of switching from oral iron–folic acid to MMN supplementation [85]. Furthermore, experts on this Delphi panel considered oral MMN supplementation to be a rational approach to treating ID/IDA in women.

As the stability and safety of modern formulations are better recognized, the use of intravenous iron for the treatment of ID/IDA has increased considerably [1,86]. Oral iron remains the cornerstone of treatment of ID/IDA in women; however, in some cases, intravenous iron is required. Intravenous iron is typically reserved for patients with severe ID/IDA and in whom oral iron is not well tolerated [26,67,87]. In addition, intravenous iron may be preferred when the rapid correction of ID/IDA is required [87]. A meta-analysis of randomized controlled trials found that while intravenous iron was more effective than oral iron in improving maternal hematological outcomes in pregnant women with IDA, the effects on neonatal hematological parameters were similar across treatment arms [88]. In a more recent clinical trial, reductions in adverse maternal and fetal outcomes were found to be similar for intravenous vs. oral iron [89]. The lack of differences in clinical outcomes vs. oral iron does not support the use of intravenous iron in preference to oral iron as the first-line treatment of ID/IDA in pregnant women [88]. Experts on the current Delphi panel concurred with the BSH guidelines which recommend intravenous iron from the second trimester onwards for pregnant women with IDA who cannot tolerate or do not respond to oral iron, and when a rapid correction of IDA is required [33]. Based on the best available evidence, experts also recommended intravenous iron in non-pregnant adult women, adolescent girls, and perimenopausal women in whom the severity of ID/IDA requires prompt management [90,91,92]. However, given the cost required for its implementation, many women in resource-limited settings still do not have access to needed intravenous iron [1].

### 4.5. Strengths and Limitations

This Delphi consensus featured several strengths. The resulting statements for which consensus was achieved reflect the views of experts with relevant expertise and experience in preventing and managing women with ID/IDA in the Asia-Pacific region. Although generated through a consensus process, these statements were derived from the medical literature and existing guidelines, and are therefore evidence-based. In addition, the high consensus threshold ensured that the recommendations were unequivocal. A few limitations should be noted. The size of the expert panel may not be representative of all Asia-Pacific countries, and therefore not generalizable to women outside of the represented countries with differing cultures and values, healthcare access, and quality of care received. For example, pregnant women in developing countries of low socioeconomic status, such as India, are less likely to receive timely iron status assessment or adhere to iron supplementation [93,94]. Secondly, the absence of online or in-person meetings after each Delphi round may have deprived panelists from exchanging pertinent information and clarifying reasons for disagreement. Additionally, it is acknowledged that the majority of statements were not supported by Level 1 studies; however, it is unlikely that large well-designed randomized controlled trials relating to the recognition, diagnosis, assessment, prevention, and treatment of ID/IDA in women will be published in the near future.

## 5. Conclusions

Within the Asia-Pacific region, obstetrician–gynecologists have a critical role to play in the identification, diagnosis, and management of ID/IDA. The primary focus should be on women at risk or presenting with ID/IDA. Consensus was also reached for 95 statements representing four domains (i.e., identification; diagnosis and assessment; prevention; and treatment of ID/IDA in women) that could be used to aid clinical decision-making for women presenting to the obstetrics and gynecology clinic and should not be interpreted as the only course of management. Key areas of consensus were around the identification and prevention of ID/IDA. There was a lack of consensus on statements relating to oral iron treatment of mild-to-moderate ID/IDA in women. In practice, patient involvement in shared decision-making around iron supplementation is key to improving adherence and clinical outcomes. Importantly, these consensus statements may serve as a pertinent step to standardizing the care of women at risk or with ID/IDA, and should be periodically reviewed to ensure consistency with the current medical literature and international guidelines.

## Figures and Tables

**Figure 1 nutrients-15-03125-f001:**
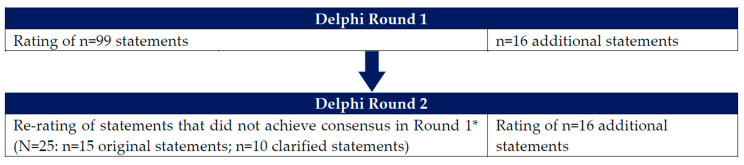
Overview of Delphi survey rounds. * Two statements that did not reach consensus in Round 1 were removed.

**Table 1 nutrients-15-03125-t001:** Overview of the Delphi survey results.

Domain	No. of Retained Delphi Items, n/N (%)	
	Round 1	Round 2
Identification of ID/IDA	22/24 (92%)	26/29 ^a^ (90%)
Diagnosis and assessment of ID/IDA	20/37 (54%)	28/35 ^b^ (80%)
Prevention of ID/IDA	12/16 (75%)	15/16 (94%)
Treatment of ID/IDA	18/22 (82%)	26/33 ^c^ (79%)
Total	72/99 (73%)	95/113 (84%)

^a^ Five new statements were added for inclusion in Round 2. ^b^ Two statements that did not reach consensus in Round 1 were removed. ^c^ 11 new statements were added for inclusion in Round 2.

**Table 2 nutrients-15-03125-t002:** Oral ferrous iron.

Formulation	Preparation	Dose (mg) *	Elemental Iron (mg) *
Ferrous fumarate	Tablet	325	106
Ferrous gluconate	Tablet	325	35
Ferrous sulphate	Tablet	325	65

* Values are based on Govindappagari, S, et al. [70].

## Data Availability

Not applicable.

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
