# Peer review of "Prevention and Management of Iron Deficiency/Iron-Deficiency Anemia in Women: An Asian Expert Consensus"

_nutrients, 2023, doi:10.3390/nu15143125_

Round 1

Reviewer 1 Report

The study found consensus on the identification and prevention of ID/IDA in women. However, there was a lack of consensus regarding the diagnostic threshold of serum ferritin during pregnancy and the role of inflammatory marker testing. The panelists unanimously agreed on prescribing oral ferrous iron as the first-line therapy for uncomplicated ID/IDA. Nevertheless, the optimal oral iron dosing for mild-to-moderate IDA remained controversial, despite a high level of consensus.

The importance of this study lies in its contribution to improving clinical practice and standardizing the care of women at risk of or presenting with ID/IDA in the Asia-Pacific region. By establishing best practices through a consensus process, healthcare professionals can have clearer guidelines and algorithms to follow when managing these conditions. This standardized approach can lead to more effective and efficient management of ID/IDA, ultimately improving the health outcomes of women in the region.

Here are a few comments:

1.     The authors need to provide more info of the key definitions, such as Delphi consensus.

2.     The title is more like review title, which is too broad. The authors need to summary the key finding and conclusions based on their data.

3.     The authors have comments on the limitations of this study. The study specifically focuses on the Asia-Pacific region, which means that the findings may not be directly applicable to other regions or populations with different healthcare systems, cultural practices, or resource availability. The generalizability of the study's recommendations may be limited. Could you give more specific examples / comments about how those above-mentioned could impact on the consensus outcomes?

4.     The Delphi consensus process involves panelists rating their level of agreement with statements. These ratings can be subjective and influenced by individual biases or personal experiences. Despite efforts to provide collective feedback and reassessment in Round 2, the initial ratings may still be influenced by individual perspectives. Could you comment on this?

The quality of English language in this passage is generally good. The sentences are well-structured, and the vocabulary used is appropriate.

Reviewer 2 Report

This study sought to determine guidance for identification, prevention and treatment of ID/ IDA in women in Asia-Pacific using a Delphi consensus process methodology.

With an expert panel of only n=8, who did not interact with each other, I was surprised with a body of evidence suggesting that the effects of ID without anemia can be detected at sFer much greater than 15 ug/L, that 15 ug/L was the cut-off that made it to the final guidance algorithms.  While these may help Clinicians make more consistent choices, do these algorithms really count as being "updated"?  Same thing goes for the Treatment -- there are so many studies citing positive treatment effects (and better compliance) with lower doses of Fe.

It was not described how the Expert Panel were chosen, nor their qualifications.  Were they aware of the aforementioned body of literature (e.g. re: sFer, among other research cited)? It was also not explained how a virtual meeting /interaction could not be managed.  This did not seem insurmountable.  
